# The Influence of Concrete Sludge from Residual Concrete on Fresh and Hardened Cement Paste Properties

**DOI:** 10.3390/ma16062531

**Published:** 2023-03-22

**Authors:** Edvinas Pocius, Džigita Nagrockienė, Ina Pundienė

**Affiliations:** Department of Building Materials and Fire Safety, Faculty of Civil Engineering, Vilnius Gediminas Technical University, Sauletekio al. 11, LT-10223 Vilnius, Lithuania; dzigita.nagrockiene@vilniustech.lt (D.N.); ina.pundiene@vilniustech.lt (I.P.)

**Keywords:** concrete sludge, residual concrete, cement paste

## Abstract

In the concrete manufacturing industry, a large amount of waste is generated. Such waste can be utilised in the production of more sustainable products with a low carbon footprint. In this study, concrete sludge, a difficult-to-utilise waste that is obtained from residual concrete by washing a concrete truck, was investigated. During washing, aggregates from the concrete mixture are separated, and the remaining insoluble fine particles combine with water to form concrete sludge. Dried and wet concrete sludge were used in the tests. Samples with different compositions were produced with dried and wet concrete sludge, cement, superplasticiser, and tap water. Seven cement pastes with different compositions were made by partially replacing cement with dried concrete sludge (0%, 5%, 10%, 15%, 20%, 25%, and 30%). In compositions with wet concrete sludge, cement was replaced by the same amounts as in the case of dried concrete sludge. The slump, setting time, and their changes with different amounts of concrete sludge were determined for fresh cement pastes. It was found that with different forms of concrete sludge, the technological properties of the mixtures change, and the setting time decreases. The density and compressive and flexural strength results were confirmed by SEM and XRD tests. The research results show that dry concrete sludge causes the deterioration of the mechanical properties of cement stone, while wet concrete sludge improves the mechanical properties of cement stone. However, it was found that replacing 5% cement with dry concrete sludge does not significantly affect the properties of hardened cement stone. In mixes with wet concrete sludge, the recommended amount of replaced cement is 10%, because the technological properties of the mixture are strongly influenced by larger amounts.

## 1. Introduction

Ecology, the environment, and sustainability are current topical issues [1]. According to the United Nations Commission on Global Application and Development, sustainability is meeting the needs of the present without compromising the ability of future generations to meet their own needs [2]. All areas of the world must inevitably become sustainable. One such area is the concrete manufacturing industry. Concrete is one of the main construction materials, and its use in the future is inevitable due to its relatively high durability, its cost, and the availability of its raw materials; the concrete industry is developed worldwide [3].

Concrete production volumes increase year by year. According to a published study, 915 million cubic metres of ready-mixed concrete was produced by nearly 30,000 concrete plants in 2018. The same study, carried out by the European Ready-Mixed Concrete Organisation, estimated that ready-mixed concrete producers in the organisation’s member countries used 260 million tonnes of cement [4]. The amount of cement used is enormous, and as a result, our environment is severely damaged, as the cement industry is one of the most polluting and carbon-intensive industries [5]. One tonne of Portland cement produced causes the emission of approx. 1 tonne of carbon dioxide [6,7,8,9,10,11,12,13,14]. Scientists have also calculated that the cement manufacturing industry emits 5% of the 30 gigatons of carbon dioxide emitted worldwide [15].

Currently, there is an active search for alternatives to natural materials used in concrete production. By-products that are obtained in other industrial areas during the production process could be used as alternatives to natural materials. Using such materials provides not only benefits to nature but also economic benefits [16,17,18]. Many researchers are looking for alternatives to cement, and the main ones being tested are fly ash, ground granulated blast furnace slag, silicon dust, and rice husks, and these are materials that are obtained from other industrial areas as by-products [19,20,21,22,23]. By using these materials and their combinations, it is possible to improve the mechanical strength and durability of cement stone [24,25].

Waste is also generated in the concrete manufacturing industry, and it is possible to use these by-products in the production of various more sustainable products with a low carbon footprint [26]. Despite the damage caused by using Portland cement, the amount of waste generated also increases with increasing production volumes. Concrete sludge is one such waste. At the end of the work day, each 9 m3 concrete mixer truck brings back approx. 200 to 400 kg of fresh concrete mix to the plant [27,28]. It has also been estimated that between 1% and 4% of the concrete mix turns into concrete sludge waste [29]. The primary source of concrete sludge is the washout water generated after flushing the chutes of ready-mixed concrete trucks at the end of the day [30]. Another source, which is particularly common in large cities, is the concrete mix returned to the mixing plant because it was not unloaded at the site due to overbooking [31]. These practices lead to the accumulation of large quantities of waste that become very costly to dispose of [32]; therefore, concrete producers need to take additional measures to tackle this problem. Residual concrete recycling systems that can separate cement particles from concrete aggregates are one of the solutions. Washed aggregates can then be used in the production of ready-mixed concrete [33]. Washout water can be passed through press filters to remove silt and obtain dry concrete sludge. Wet concrete sludge is obtained without filtering the washout water.

Most of the research into concrete waste has been performed with recycled aggregates and their use in the production of ready-mixed concrete, whereas particles with sizes below 5 mm have been researched less [34]. Some researchers [35] studied the use of washed sand from residual concrete recycling systems. They observed that washed sand was contaminated with fine cement particles, which affect the particle size distribution and the bulk density of the sand. Such sand can be used in concrete mixes, but higher amounts of superplasticisers must be added.

Other researchers received positive results by using concrete returned to the batch plant as an aggregate [36]. They added non-toxic chemical additives to a not-yet-hardened concrete mixture and turned it into a granulated material, which was later used as concrete aggregate.

Other researchers tried to use concrete sludge in the manufacture of precast concrete blocks. They used dry concrete sludge and washed sand from residual concrete processing systems and hardened the obtained composite cementitious material in a CO2 environment. Tests showed that the wall blocks made from this mix were carbon-neutral and sustainable.

There were attempts to use concrete sludge from batching plants as a raw material in cement manufacturing [37]. The tests revealed that it is impossible to use concrete sludge as a raw material in cement manufacturing.

Researchers also tried to use concrete sludge as a microfiller for mortars [38]. They found that dry concrete sludge had a negative effect on the workability of the mortars and required much higher amounts of superplasticisers. A high variation in compressive strength from −30% to +17% among the specimens in comparison to the control specimen was another drawback.

Researchers also tried to use concrete sludge instead of cement to stabilise road foundation layers [39]. Positive results of using concrete sludge to stabilise foundations were reported [40]. The results showed that, in terms of compressive strength, it was possible to achieve 38% of the stabilisation effect compared to the same amount of standard Portland cement.

Researchers [41] tested in detail the concrete sludge obtained from local batching plants. They found that cement particles and a small amount of sand fines prevailed in concrete sludge. The tested concrete sludge contained 49.7% cement particles, 38.7% sand fines, and 11.6% limestone filler.

This paper analyses the effects of wet concrete sludge and dry concrete sludge on the properties of hardened cement paste. The chemical and mineral compositions of the sludge were analysed. The thermal stability of the material and its fraction of volatile components were determined by means of thermogravimetric analysis. The slump flow and the setting time were determined for the cement paste, and the density and compressive and flexural strengths were determined for the hardened cement paste.

## 2. Materials and Methods

### 2.1. Raw Materials

Cement CEM I 42.5 R that meets the requirements of LST EN 197-1 [42] and concrete sludge obtained from residual concrete processing systems after the separation of coarse aggregates were used for the tests. Both wet and dry concrete sludge were tested. For dry concrete sludge testing, wet concrete sludge was dried at a 120 ± 5 °C temperature to a constant mass. Table 1 illustrates the chemical composition of dry concrete sludge, and Table 2 illustrates the physical properties of the sludge.

The density and dry solids content of wet concrete sludge were determined. This was carried out according to LST EN 1008-2005 [43].

The density of concrete sludge used for the tests was 1220 kg/m^3^. The dry solids content of the sludge was 500 kg/m^3^. The pH of the wet sludge, determined according to LST ISO 4316:1997 [44], was 11.4, and the cement paste pH value was 14 when the w/c ratio in the cement paste was 0.35.

The particle size distribution analysis of dry concrete sludge revealed that particles with an average size of 15.85 µm prevailed in the tested sludge. At 10%, the particles were 3.35 μm; at 50%, they were 10.18 μm; and at 90%, they were 12.88 μm. The size of CEM I 42.5 R particles ranged from 0.1 to 70 μm. The particle diameter was 0.75 μm at 10%, 7.88 μm at 50%, and 39.66 μm at 90%. The mean particle diameter was 14.21 μm. Figure 1 shows the particle distributions of the dried concrete sludge and used cement. From our results, we can see that Portland cement has more fine particles than dried concrete sludge, and the particles are more evenly distributed compared to those of used cement.

### 2.2. Paste Design and Sample Preparation

The cement pastes were prepared according to the mix design in Table 3. Two different mixes were made. The first mix was prepared to study the effect of dried concrete sludge on fresh and hardened cement pastes. In the first mix, cement was replaced by dried concrete sludge at 0%, 5%, 10%, 15%, 20%, 25%, and 30%. The second mix contained wet concrete sludge, which replaced some of the cement. The amount of added wet concrete sludge was two times higher than that of the dried concrete sludge, because the wet concrete sludge contains 2 times lower dry solids content than dried concrete sludge. Cement was replaced by the same amounts as in the first mix with dried concrete sludge (5%, 10%, 15%, 20%, 25%, and 30%). In both mixes, w/c was kept the same (0.35), as was the chemical admixture (0.40%).

After mixing, the fresh pastes were cast in 160 × 40 × 40 mm steel moulds and compacted on a vibrating table for 20 s. After compacting, all specimens were left for 12 h at an ambient temperature of 20 ± 2 °C and then demoulded and placed under water at a temperature of 20 ± 2 °C. All specimens were cured under water for 28 days.

### 2.3. Test Methods

The dry method was used to determine the specific surface area and particle size distribution of dry concrete sludge particles using the particle size analyser Cilas 1090 LD in the interval from 0.01 µm to 500 µm using air as a carrier. The particles were dispersed by ultrasound until a 12% distribution of the material in the media was reached. The measurement span was 60 s. A standard operating system (Fraunhöfer) was used.

The XRD analysis was conducted with an X-ray diffractometer (BRUKER AXS D8 ADVANCE, Bruker Corporatio, Rheinstetten, Germany). The following XRD parameters were used: CuKα radiation, Ni filter, detector step of 0.02°, intensity measuring span of 0.5 s, anode voltage Ua = 40 kV, and current I = 40 mA. The accuracy of XRD measurements was 2θ = 0.01°.

X-ray fluorescence spectroscopy was performed with a spectrometer (Bruker X-ray S8 Tiger WD). A Rh target X-ray tube was used with an anode voltage of up to 60 kV and a current (I) of up to 130 mA. The specimens were measured in a helium atmosphere. The SPECTRA Plus QUANT EXPRESS method was used for the measurements. The microstructures of the materials were observed with a scanning electron microscopy (SEM) device (SEM JEOL JSM-7600F, JEOL (Germany) GmbH, Freising, Germany).

The compositions of hardened cement pastes are presented in Table 3. In the test mixtures with wet sludge, the amount of cement and tap water was reduced by replacing it with wet concrete sludge, whereas in the test mixtures with dry concrete sludge, the amount of cement was reduced by replacing it with concrete sludge.

The main properties of concrete were determined according to the following standards: LST EN 12350-5:2019 for the slump flow [45], LST EN 12390-7:2019 for the density of hardened cement paste [46], LST EN 12390-3:2019 for compressive strength [47], and LST EN 12390-5:2019 for flexural strength [48].

## 3. Results and Discussion

### 3.1. Parameters of Concrete Sludge

The following materials were identified in the XRD spectra of dry concrete sludge: 36% calcite, 16% portlandite, 14% dolomite, 11% quartz, 7.6% kuzelite, and 3.8% ettringite. Figure 2 illustrates the XRD image of dry concrete sludge.

The microstructure of concrete sludge was observed using the Helios NanoLab 650 SEM instrument. The obtained physical property results of dry concrete sludge are presented in Table 2. The images of the microstructure show that the dry concrete sludge is made of round-shaped crystals, which are in calcite and hexagonal plates. Hexagonal plates are portlandite crystals, which are large in size, reaching 60–100 μm. There is some research [49,50,51] indicating that portlandite crystals can reach up to 100 microns in size if cement pastes have a high water-to-cement ratio. When the water-to-cement ratio is lower than 0.25, portlandite crystals appear in nanometric dimensions and are dispersed in the C-S-H gel. SEM studies confirm that a large amount of water in wet concrete sludge leads to the formation of large-sized portlandite crystals.

Thermogravimetric analysis was carried out for dry concrete sludge. The analysis results revealed an intensive endothermic effect and mass loss of about 13% at a temperature between 100 °C and 200 °C. These results can be attributed to the loss of hydrated water in cement minerals and the decomposition of ettringite and monocarboaluminate. The decomposition of ettringite, with about 9% weight loss, shows an endothermic effect in the temperature range of 100–140 °C. Such results confirm the XRD data in which the presence of ettringite and kuzelite is identified (Figure 3). However, the fact that the dry concrete sludge sample was dried at a 120°C temperature may explain why no ettringite was found in the SEM image (Figure 4). A second small endothermic effect with about 3% weight loss occurs in the temperature range between 158 and 175 °C and is attributed to monocarboaluminate decomposition [52]. In the temperature interval between 450 °C and 550 °C, portlandite decomposition is observed. The third endothermic effect occurs in the temperature range between 690 °C and 790 °C, where the decomposition of calcite occurs. The total mass loss of the sample is 29.5%.

### 3.2. Properties of Fresh Cement Paste

The slump flow tests of cement paste are presented in Figure 5 and it showed that the slump increases when cement is replaced with dry concrete sludge. The greater the amount of cement replaced with dry concrete sludge, the higher the slump flow. The slump flow of the specimens in which 30% of cement was replaced with dry concrete sludge increased by 56.7% compared to the control specimen. The opposite trend was observed in the specimens with wet concrete sludge. When a larger amount of cement was replaced with wet concrete sludge, the slump flow was lower, because the density of wet sludge is higher than that of water, and the viscosity of the paste increased. When the amount of cement replaced by wet concrete sludge increased to 30%, the slump flow was reduced by 45.3%.

The initial setting time of the specimens with both dry and wet concrete sludge decreased with a higher sludge content in the specimens. It was observed that the change in the initial setting time was very similar irrespective of the form of concrete sludge added to the mixtures. This can be explained by the presence of a significant amount of calcite in the sludge, as calcite is known to decrease the setting time [53]. A decrease in setting time can also be explained by calcium hydroxide and ettringite, which were found in concrete sludge. Results of the initial setting time is presented in the Figure 6.

### 3.3. Hardened Cement Pastes’ Properties

To determine the difference in the chemical compositions of samples without concrete sludge and with dried and wet concrete sludge, XRD analysis was performed. Specimens with no additive, with 10% dry concrete sludge, and with 10% wet concrete sludge were tested. The tests showed that all the samples consist of portlandite, calcite, dolomite, ettringite, and CSH. Only the amounts of these compounds in the samples differ. The results are shown in Figure 7.

The amount of portlandite in all of the samples, despite different cement contents, is almost the same, but with a tendency to increase in compositions with sludge, and it is around 41–45%. This may mean that portlandite from sludge can additionally contribute to the portlandite amount in a sample. However, bearing in mind that the amount of portlandite, according to the results of X-ray analysis calculations for dry sludge (Figure 3), is 16%, the sludge (dry or wet) intensified cement mineral hydration.

In different samples, similar amounts of calcite were found: 20.3% with dried concrete sludge, 18.1% with wet concrete sludge, and 23.0% in the control sample. These results show that in the sample with wet concrete sludge, more calcium ions participate in the hydration products’ crystallisation process. Different amounts of non-reacted C_3_S and C_2_S were detected; in the first sample, without concrete sludge, there was 15.4%; in the second sample, with dried concrete sludge, there was 9.5%; and in the third sample, with wet concrete sludge, there was 5.6%. This decrease in C_2_S and C_3_S is obvious, because the amount of cement was decreased in the samples with sludge. However, in the sample with wet concrete sludge, unreacted C_2_S and C_3_S is almost 1.7 times lower than in the sample with dry concrete sludge. This result shows that the hydration of cement minerals in the presence of wet sludge takes place much more actively. The amounts of ettringite are also different. The amounts of ettringite for the same time CSH-type mineral amounts in the samples with concrete sludge are different: 2.7% in the sample with dried concrete sludge, 5.9% in the sample with wet concrete sludge, and 3.2% in the control sample. This may mean that wet concrete sludge promotes CSH-type mineral crystallisation. It seems that wet conditions in the wet sludge promote more active cement mineral hydration, which may be due to a specific layer of water covering sludge particles. In the control sample, no quartz is obtained. In the samples with sludge, some amount of quartz is identified. Dolomite is obtained in all three samples, but in the samples with sludge, the dolomite amount is about 3 times higher than in the control sample.

Figure 8 shows the samples’ microstructure images at different magnifications. It is obvious that the concrete sludge (wet or dry) thickens the microstructure of the sample, apparently due to the larger amounts of hydration products.

Density tests on the specimens with dry and wet cement sludge (Figure 9) showed the opposite tendency. The density of the specimens in which cement was replaced with dry concrete sludge gradually decreased, whereas the density of the specimens in which cement was replaced with wet concrete sludge increased. As the bulk density of dry concrete sludge is lower than the density of cement, the density of hardened cement paste may decrease. The density of the specimens in which 30% of cement was replaced with dry concrete sludge decreased by 90 kg/m^3^ in comparison to the control specimen, whereas the density of the specimens in which 30% of cement was replaced with wet concrete sludge increased by 88 kg/m^3^. This could have happened due to the wet concrete sludge’s density; because the sludge is denser (1220 kg/m^3^) than the used water, a lower slump flow also creates denser microstructures of specimens. Additionally, it can be noticed that the amount of ettringite is lower in specimens with wet sludge than in specimens with dry concrete sludge.

As with the density results, the strength results revealed different trends in the moulded specimens containing dry and wet concrete sludge. These trends are visible in Figure 10. The compressive strength at 28 days decreased in parallel with the higher content of dry concrete sludge replacing cement in the specimens.

There was a slight drop in compressive strength when 10% of cement was replaced with dry concrete sludge. In comparison to the control specimen, the compressive strength was reduced by 3.6 MPa. In the specimens in which 30% of cement was replaced with dry concrete sludge, the compressive strength was reduced by 19.5 MPa. On the contrary, in the specimens in which wet concrete sludge replaced different amounts of cement, the compressive strength increased. When 60% of wet concrete sludge (or 30% of dry matter) was added, the compressive strength increased by 19.7 MPa compared to the control specimen.

The same trend was observed for the flexural strength from the test results illustrated in Figure 11. When 30% of cement was replaced with dry concrete sludge, the flexural strength was reduced by 34.3% compared to the control specimen, but when only 10% of cement was replaced, the strength was reduced by 4.4%. The flexural strength increased by 31.8% when 60% of wet concrete sludge (or 30% of dry matter) was used.

Both the compressive strength and the flexural strength of the specimens containing dry concrete sludge decreased significantly, presumably due to the dilution effect [54] and more porous structure, because the density of the sample decreases. Potentially large portlandite crystals create a more porous structure around themselves when cement is replaced with dry concrete sludge. In addition, CSH-type compounds are also produced in small amounts in this specimen. The strength of the specimens with wet concrete sludge increased significantly, presumably due to the denser structure and the increased density of specimens, and due to the specific wet conditions of the sludge, the layer of water covering sludge particles speeds up the hydration of cement minerals and specimens, achieving greater strength. In addition, XRD studies (Figure 7) show that this specimen contains a 2 times higher amount of CSH than the specimen with dry concrete sludge.

## 4. Conclusions

In this research, the possibility of partial cement replacement by wet and dry concrete sludge was studied. The slump flow, setting time, created structure, and physical–mechanical properties of cement stone have been analysed. The following conclusions were made after conducting the research.

The flowability of the cement paste increased in proportion to the amount of cement replaced with dry concrete sludge. When 30% of cement was replaced with dry concrete sludge, the flow rate increased by 56.7%. In cement paste with wet concrete sludge, on the contrary, the flowability of the cement paste decreased. When 30% of cement was replaced by wet concrete sludge, the flowability of concrete was reduced by 45.3%.

The results of the setting time in the pastes with dry and wet concrete sludge showed that both dry and wet concrete sludge shortened the setting time. The greater the amount of concrete sludge added, the shorter the setting time. Compared to the control specimens, the setting time of the specimens modified with dry concrete sludge was shortened by 56 min, and in the specimens with wet concrete sludge, the setting time was shortened by 46 min.

XRD analysis shows that portlandite from sludge increased the portlandite amount in specimens and participated in hydration processes. In the sample where cement was replaced with wet concrete sludge, unreacted C_2_S and C_3_S amounts were 1.7 times lower than in the sample where cement was replaced with dry concrete sludge, but the amounts of CSH-type compounds were 2 times higher. This result shows that wet concrete sludge promotes the hydration of cement and CSH-type mineral crystallisation.

The replacement of cement by dry concrete sludge decreased the density by 4.1% when 30% of cement was replaced in comparison to the control specimen, but the replacement of 30% cement by wet concrete sludge, due to the higher sludge density, increased the specimen density by 4%. In comparison to the control specimen, increasing the replacement of cement by dry concrete sludge decreased the compressive strength of the specimen by up to 32.7%, whereas increasing the replacement of cement by wet concrete sludge increased the compressive strength of the specimen by up to 1.32 times. The strength of the specimens with wet concrete sludge increased due to the denser structure and the promoted hydration of cement. The results of the flexural strength indicate the same trend as was observed for the compressive strength.

It was found that a small amount of cement, up to 5%, can be replaced with dry concrete sludge, whereas the optimal amount of cement replaced by wet sludge is 10%.

## Figures and Tables

**Figure 1 materials-16-02531-f001:**
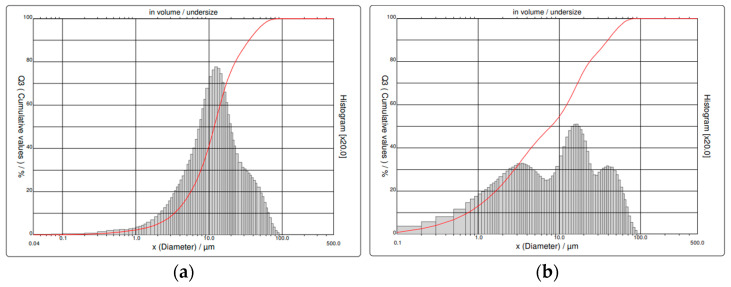
Particle distributions of (**a**) dried concrete sludge and (**b**) cement.

**Figure 2 materials-16-02531-f002:**
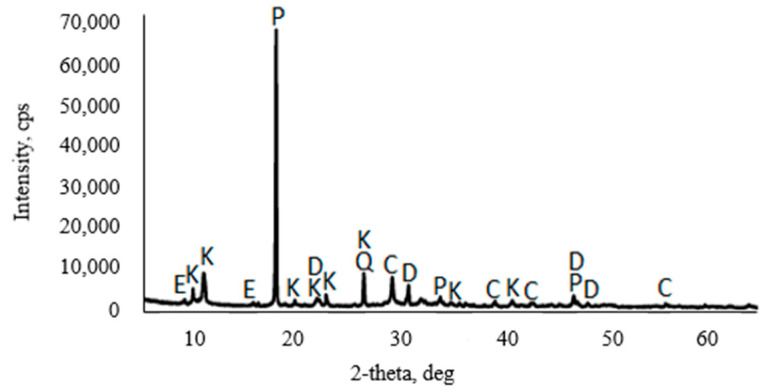
XRD image of dry concrete sludge: C—calcite; P—portlandite; D—dolomite; Q—quartz; E—ettringite; K—kuzelite.

**Figure 3 materials-16-02531-f003:**
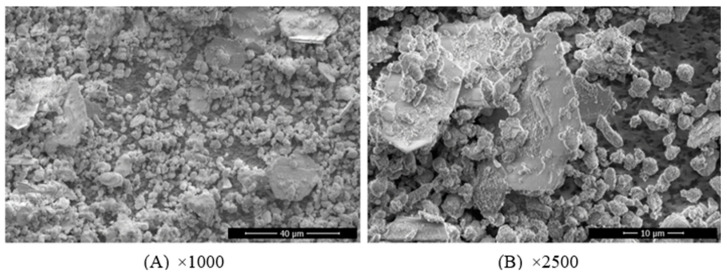
Microstructure of dry concrete sludge: (**A**) ×1000 magnification and (**B**) ×2500 magnification.

**Figure 4 materials-16-02531-f004:**
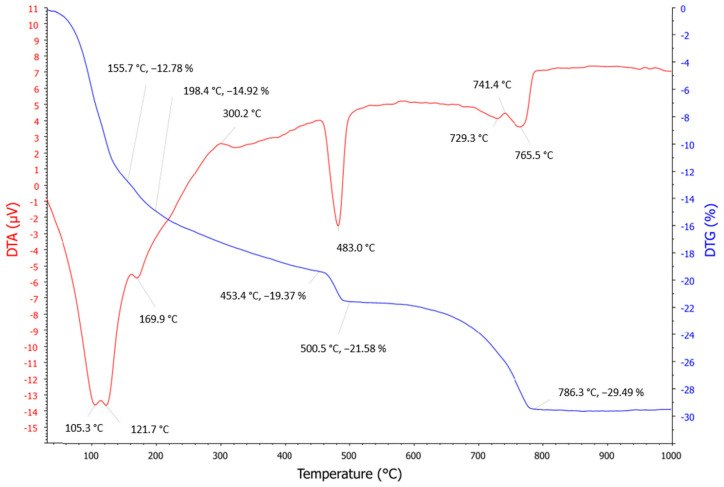
TGA graph of dry concrete sludge.

**Figure 5 materials-16-02531-f005:**
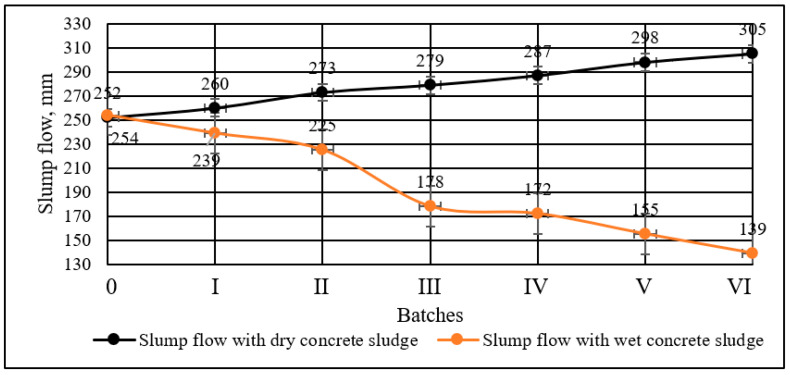
Slump flow of cement paste with different amounts of cement replaced with wet and dry concrete sludge.

**Figure 6 materials-16-02531-f006:**
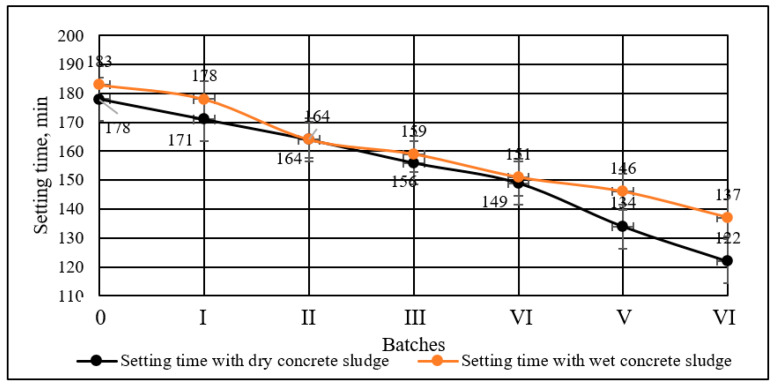
Initial setting time of cement paste with different amounts of cement replaced with wet and dry concrete sludge.

**Figure 7 materials-16-02531-f007:**
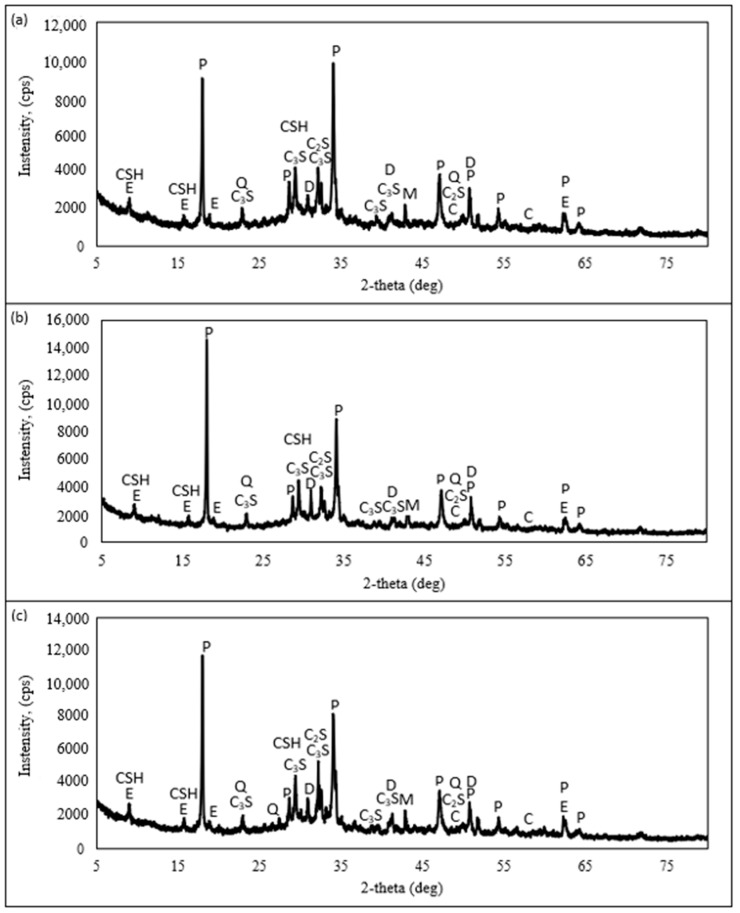
XRD image of hardened cement paste: (**a**) without sludge, (**b**) with 10% cement replaced with dry concrete sludge, (**c**) with 10% cement replaced with wet concrete sludge; C—calcite; P—portlandite; D—dolomite; E—ettringite; C_3_S—tricalcium silicate; C_2_S—dicalcium silicate; CHS—calcium silicate hydrate; Q—quartz.

**Figure 8 materials-16-02531-f008:**
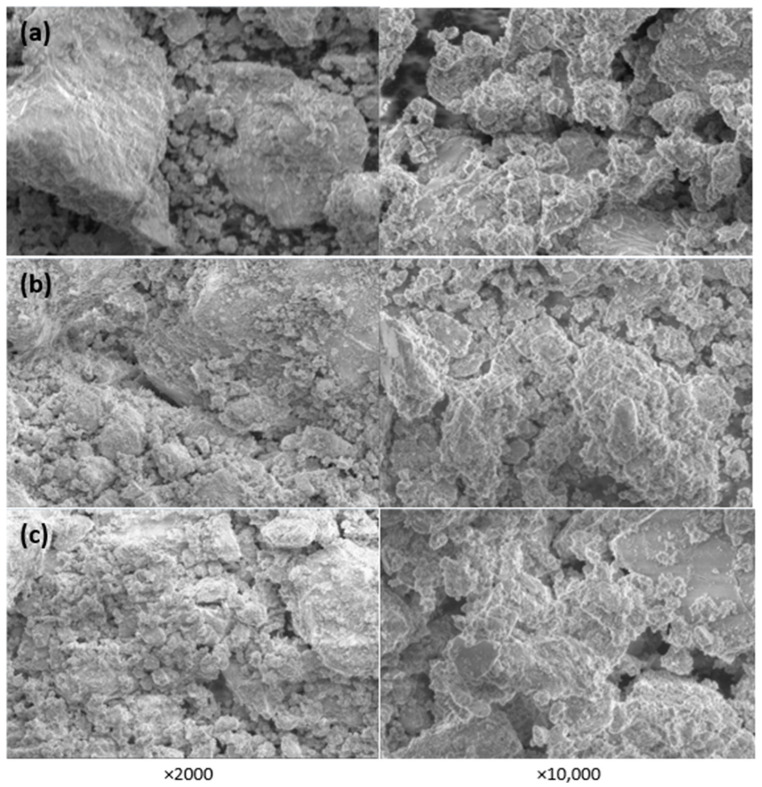
SEM images of cement stone: (**a**) with no concrete sludge, (**b**) with 10% cement replaced with dry concrete sludge, (**c**) with 10% cement replaced with wet concrete sludge.

**Figure 9 materials-16-02531-f009:**
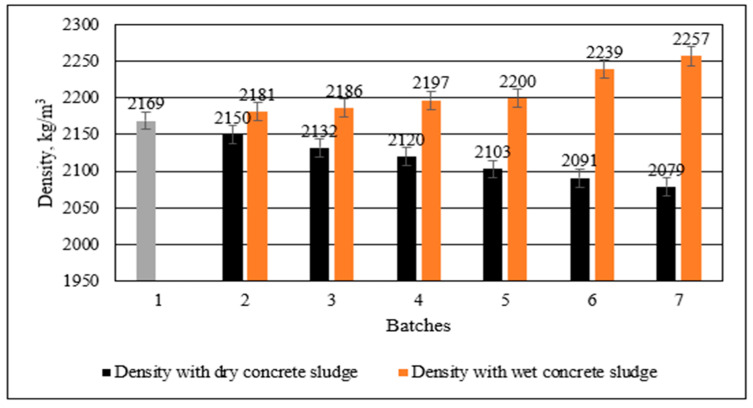
Density of concrete with different amounts of cement replaced with wet and dry concrete sludge.

**Figure 10 materials-16-02531-f010:**
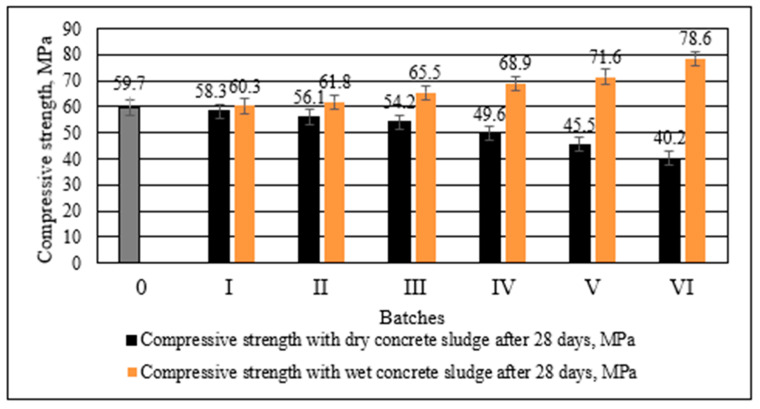
Compressive strength of concrete with different amounts of cement replaced with wet and dry concrete sludge after 28 days.

**Figure 11 materials-16-02531-f011:**
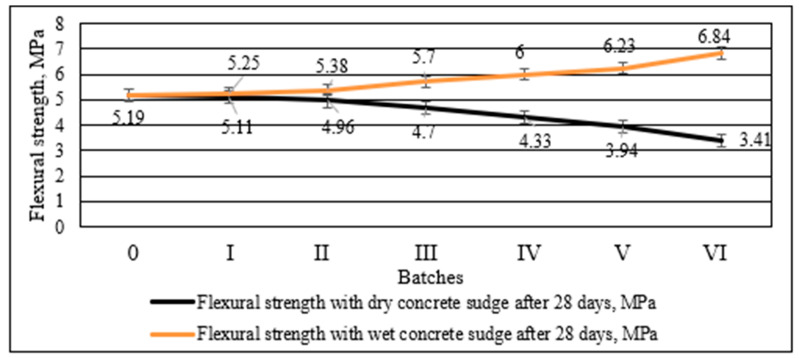
Flexural strength of concrete with different amounts of cement replacement with wet and dry concrete sludge after 28 days.

**Table 1 materials-16-02531-t001:** Chemical composition of dry concrete sludge.

Chemical Composition of Concrete Sludge, %											
CaO	SiO_2_	Al_2_O_3_	SO_3_	Fe_2_O_3_	MgO	K_2_O	P_2_O_5_	TiO_2_	SrO	MnO	Cl
40.9	14.9	3.14	2.40	2.37	2.26	0.89	0.38	0.21	0.06	0.04	0.03

**Table 2 materials-16-02531-t002:** Properties of dry concrete sludge.

Properties	Dry Concrete Sludge
Specific surface area, cm^2^/g	316
Particle density, kg/m^3^	2774
Bulk density, kg/m^3^	826

**Table 3 materials-16-02531-t003:** Mixing proportion of hardened cement paste.

**With Dried Concrete Sludge**							
Batches	0	I	II	III	IV	V	VI
Cement, %	100	95	90	85	80	75	70
Water, %	100						
Chemical admixture, %	0.40						
Dry concrete sludge, %	0	5	10	15	20	25	30
w/c	0.35						
**With Wet Concrete Sludge**							
Cement, %	100	95	90	85	80	75	70
Water, %	100	95	90	85	80	75	70
Chemical admixture, %	0.40						
Wet concrete sludge, %	0	10	20	30 430	40	50	60
w/c				0.35			

## Data Availability

Not applicable.

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
