# Peer review of "The Influence of Concrete Sludge from Residual Concrete on Fresh and Hardened Cement Paste Properties"

_materials, 2023, doi:10.3390/ma16062531_

Round 1

Reviewer 1 Report

The research in this manuscript is of great significance and enriches the research on recycled concrete. This manuscript is basically at publishable level, but it is recommended to further optimize the logical structure of the abstract, introduction and conclusion before publication. In addition, the authors need to add some pictures of the experimental process to further improve the completeness and credibility of the manuscript.

Author Response

I wanted to thank you for your review. It means a lot that you acknowledged the significance of our research topic. We will take into consideration about your comments further optimization of the logical structure of the abstract, introduction and conclusion. Maybe you have some insights where and how exactly could it be done? Your observation, that some pictures could be added got us thinking what pictures we could use from the experimental process, because it is not a lot of experiments was done, where photos could be taken and the images would show results and could bring clarity. If you have any advice on what pictures could be used to bring clarity, please specify. All in all, again we wanted to thank you for your review.

Reviewer 2 Report

The paper presents a study of the properties of a cement paste, where parts of the cement have been replaced by dry or wet concrete sludge. Concrete sludge was explained as the remaining fine particles (and water) after washing left-over concrete from concrete mixers and trucks. The study showed that higher levels of wet sludge reduced the slump flow and the setting time, while it increased the density and the compressive strength. For increasing levels of dry sludge, the slump flow increased while the setting time decreased, and the density as well as the compressive strength decreased.

The paper presents a very interesting topic, dealing with sustainability issues and trying to reduce the waste of concrete production. The background, methods and results are relatively well presented and support the conclusions of the study. However, some clarifications are needed.

At first I was a bit confused when you were discussing concrete production and concrete sludge, but what you studied was a cement paste. The title specifies that you were looking at a cement paste, but perhaps you should have some early clarification in the text also to avoid confusion by other readers (or maybe it was just me?).

The paper keeps talking about technological properties of the cement paste, but I would prefer using the term Material properties instead, since it is a better explanation to what you are actually studying.

Some comments to the methods chapter. In line 123 you give an unreasonable low density of the sludge 1.22 kg/m3. I would like to have a better explanation of the concrete sludge, i.e. how did you produce it, was it the same concrete mix (without aggregates), did you use the same cement, what was the w/c ratio for the sludge, did you use the same sludge-batch for all tests, etc? On line 152 and in table 3 you mention s/c, what is this? Perhaps it is the w/c which you don’t mention. I would also like a clearer explanation on how you accounted for the extra water from the wet sludge since you did not change the w/c ratio.

Some comments for the results and conclusions. I don’t get the same percent-numbers as you when I use the numbers you have in your results. For example, when the wet sludge increased to 30% you state that the slump flow reduced by 82%. Initially you had a slump flow of 252 mm, which implies the slump flow for 30% sludge would be 0.18 x 252 = 45 mm. I saw more errors in the percentage numbers presented in the paper and all these calculations needs to be checked.

Overall I liked the paper and my recommendation is to accept the paper after a minor revision, including a small language check.

Congratulations to the authors to an interesting study!

Author Response

First, we wanted to thank you for your very detailed review and observations. Also thank you for acknowledgement of our research, that is interesting and have significance when dealing with the sustainability issues and trying to reduce the wastes of concrete production.

Secondly, in the beginning we wanted shortly discuss about concrete production and concrete sludge because it is precisely the concrete production industry that faces problems related to these wastes. Also we think it is important to discuss how and from what industrial sector these wastes are generated. We really didn‘t mean it to be confusing, we just wanted to bring more clarity.

Third, your observation about talking about technological properties. We agree it is not the best term to use when talking cement paste. After your remarks we decided to change the term to make it clearer what was studied and to use the term rheological properties.

Forth. Thank you for spotting the error about sludge density. Due to clerical error and units of measurement, the density was incorrect. According to the specified measurement units, it should be 1220 kg/m3. To explain how the sludge is produced, it was gathered from the concrete mixing plant where the sludge is generated. It is generated from washing all kind of concrete mixes, aggregates are washed out and in the sludge only remain fine particles. Sludge was not produced by us on purpose, so we did not use cement and water cement ratio is unknown. For the tests the sludge was used the same, from one take. Further addressing your observation about s/c, you are right, that there should be w/c and this is a mistype and it will be corrected. We maintained the same w/c ratio by calculating how much water is in the sludge. We calculated water content in the sludge using sludge density and density of fine particles.

Fifth, thank you for spotting the mistakes in calculating the percentages. We will check all the calculations and will correct them. 

As a final point, we want to thank you for very detailed review, given comments and noticed errors.

Reviewer 3 Report

I congratulate the authors of a very interesting work on the possibility of reducing the carbon footprint in the concrete mix by using production waste in the form of sludge from the production of concrete in dry and wet form. The test results confirm the possibility of using wet sludge, which not only significantly improves the wall strength, but also significantly increases the workability of the mixture over time.

An additional important argument supporting the legitimacy of the research undertaken is the possibility of a significant reduction of the carbon footprint in concrete through the use of recycling waste in the form of wet sludge.

The research is interesting, but I have a question for the authors, how do they industrially predict the possibility of using this type of additive. How will it be put into production and how will the weighing process be controlled?

Author Response

First of all thank you for your review. We are happy to see that you agree with us that our research could be helpful in reducing carbon footprint in concrete production. It is important to use production waste in production of fresh concrete mix to make concrete more friendly to the environment.

Secondly, thank you for your observation about using recycling waste in the form of wet sludge. It is a great significance for us to investigate wet sludge and how it effects material properties and that we got good results with it.

Third, thank you for the good questions. About the use of this additive industrially is hard to predict. We think it is truly possible, because a lot of concrete mix plants have these wastes and it costs additional money to utilize it. It could help industry to save money, to make their product have lower carbon footprint and to ensure circular economy. It can be put into production by using necessary equipment where to wash the concrete leftovers, tanks, where sludge can be stored, equipment to pump it. It could be weighed together with water and then dosed into the mixer.

In summary, I wanted to thank you again for your review, observations and interesting questions.
